# Analysis of Critical Control Points of Post-Harvest Diseases in the Material Flow of Nam Dok Mai Mango Exported to Japan

**Benjamaporn Matulaprungsan [1]**, **Chalermchai Wongs-Aree [1,2,*]**, **Pathompong Penchaiya [2]**, **Panida Boonyaritthongchai [1,2]**, **Viroat Srisurapanon [3]** and **Sirichai Kanlayanarat [2]**

1   Postharvest Technology Program, School of Bioresources and Technology, King Mongkut's University of Technology Thonburi, Bangkok 10150, Thailand
2   Postharvest Technology Innovation Center, Office of the Higher Education Commission, Bangkok 10400, Thailand
3   Department of Civil Engineering, Faculty of Engineering, King Mongkut's University of Technology Thonburi, Bangkok 10140, Thailand
*   Correspondence: chalermchai.won@kmutt.ac.th; Tel.: +66-2470-7725

**Abstract:** 'Nam Dok Mai' mango is a luxury commercial fruit in Thailand, but post-harvest diseases infecting the ripe fruit is a major problem affecting marketability. The objective of the present study was to map the supply chain of 'Nam Dok Mai' mangoes exported to Japan and analyze the critical points of post-harvest disease infection caused mainly by *Colletotrichum gloeosporioides*. Risk points of the post-harvest diseases were found by examining the material and information flows from processes ranging from field production to post-harvest handling, and these were obtained from mango growers and an exporter. The findings of interviews with mango growers and observations of the mangoes in field production were that the first point of risk was cultivar selection, while branch pruning and fruit bagging were further important processes causing post-harvest fruit decay. On the other hand, it was found that post-harvest handling was significant in decreasing anthracnose disease infection; this was seen at the step of dipping the fruit in 50 °C hot water for 3 min at the processing line.

**Keywords:** material flow; information flow; exporter; mango growers; risk points; post-harvest diseases

## 1. Introduction

'Nam Dok Mai' mango (*Mangifera indica* Linn) is one of Thailand's most economically important fruits due to its flavor and bright yellow flesh, which is favored by consumers in both domestic and export markets. The major production areas of 'Nam Dok Mai' mango are located in the Chiang Mai province in the north, the Phitsanulok province in central Thailand, the Loei and Nakhon Ratchasima provinces in the northeast, and the Prachuap Khiri Khan province in the south [1]. The on-season production of mango in Thailand occurs from April to May and it extends to late July in some regions. Early off-season production occurs from January to March and late off-season production occurs from August to December [2]. The export volume of fresh mangoes in 2017 was 55,792 tons, which was valued at 70 million USD, with Nam Dok Mai being the main cultivar [1]. Thailand exports mangoes mainly to Japan, Korea, Vietnam, China, and Malaysia [2]. 'Nam Dok Mai' mango is a preferred product in many countries due to favorable characteristics such as attractive skin color, pulp color, taste, and flavor.

However, a crucial problem in terms of 'Nam Dok Mai' mango production for export is the post-harvest disease infection of anthracnose and stem end rot diseases caused by *Colletotrichum gloeosporioides* and *Lasiodiplodia theobromae*, respectively. *C. gloeosporioides* is the most significant fungus

that can infect all parts of a mango tree including the stem, leaf, flower, immature fruit, and mature fruit [3]. Consequently, anthracnose as a latent infection may enter the mango during the immature stage or even during flowering [4,5]. The symptoms of the disease do not express themselves in an infected unripe mango but appear when the mango ripens. The symptoms were seen to be black spots or flecks on the skin of ripe mangoes. In recent years, researchers have examined the control of the anthracnose disease during preharvest and post-harvest processes such as chemical use [6], biocontrol [7], cultivation [8], and so on.

In practice, mango production processes both at the preharvest (at the field of production) and post-harvest (at the packing house) stages had different regulations and protocols for certification. These led us to investigate the problem through the material and information flows obtained from the mango production process, ranging from up-stream to down-stream processes, and to look for risk points for post-harvest disease control. The examination of materials and information flows during production processes could be a powerful tool to determine the traceability of 'Nam Dok Mai' mango production system. Our study focused on mangoes exported to Japan in particular. The fruit had to be produced under specific conditions approved by Good Agricultural Practices (GAP) in the field, and as per a special requirement, they had to be treated using vapor heat treatment (VHT) to increase the pulp temperature to 47 °C constant for 20 min during post-harvest handling [9]. Furthermore, the maximum residue limits (MRL) of Chlorpyrifos, an important pesticide, on the fruit had to be less than 0.05 mg·L$^{-1}$ [10].

Supply chain management (SCM) is a processing alignment of material, financial, and information flows to improve activities and processes. SCM involves the management of relationships between efficient production and the supply of products from farms to customers [11]. To evaluate SCM, the supply chain operation reference (SCOR) model, which provides a framework of business processes, practices, and technology design [12,13], has been used as a strategic tool. This study focused on mapping the material flow and information flow of mangoes exported to Japan. The SCOR model of the mango supply chain was monitored. Diseases infecting mangoes during post-harvest were monitored and the risk points of diseases infecting mangoes during the mango supply chain were critically analyzed. This study resulted in certain critical findings regarding the risks of disease infection during mango processing and mango export both at the preharvest and post-harvest stages.

## 2. Materials and Methods

The material and information flows in the supply chain of 'Nam Dok Mai' mango for export to Japan were collected from mango growers and an export company in 2016–2017. The data of post-harvest disease infection during the supply chain of 'Nam Dok Mai' mango was identified using observation, questionnaires, and interviews with growers and exporter staff. This study adapted the method from Olsen and Aschan (2010) [14] and the processed mapping of mango exported to Japan is shown in Figure 1.

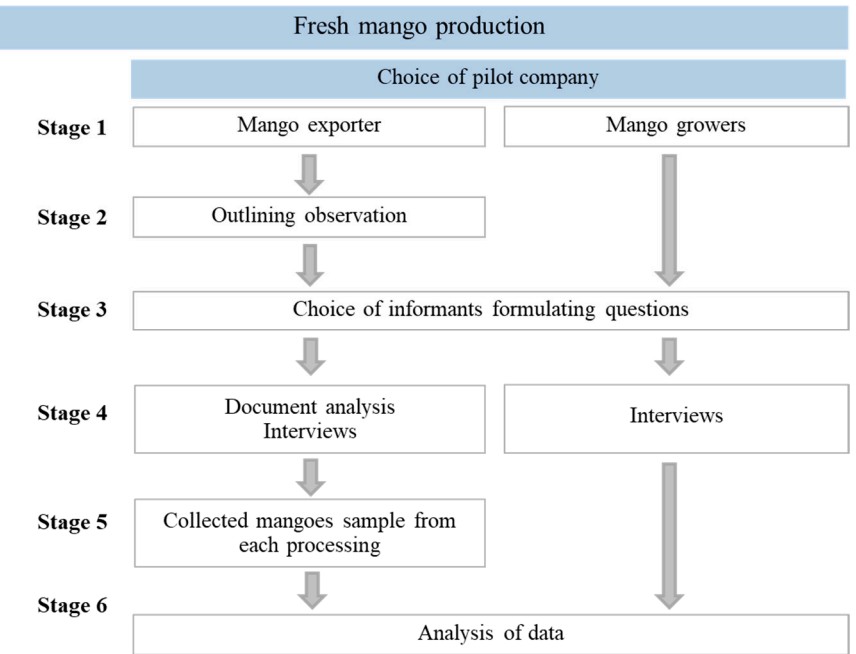

**Figure 1.** Overview of steps in the process mapping of the production of 'Nam Dok Mai' mango exported to Japan.

## 2.1. Choices of Pilot Company (Stage 1)

A pilot company located at the Rojana Industrial Park in the Ayutthaya province, Central Thailand (an hour from the center of Bangkok), which exports 'Nam Dok Mai' mango to Japan, was selected and 44 growers who supply mangoes to the company from different production locations across the country were chosen.

## 2.2. Outlining Observation (Stage 2)

The exporter was interviewed to gain information regarding the material flow procedure of post-harvest mango handling. The problems of barriers during management were also recorded.

## 2.3. Choices of Informants and Formulating Questions (Stage 3)

The primary problem identified by the mango exporter was post-harvest disease. Therefore, the questionnaire was based on preharvest and post-harvest mango management following mango cultivation processes and the Thailand GAP guide.

## 2.4. Interviews and Document Analysis (Stage 4)

### 2.4.1. Mango Growers

Mango growers who supply 'Nam Dok Mai' mango to the company were chosen from 4 different locations and the list included 12 growers from the Phetchabun province (north), 11 growers from the Nakhon Ratchasima province (northeast), 11 growers from the Sa Kaeo province (east), and 10 growers from the Prachuap Khiri Khan province (south). All growers were classified into 3 groups based on farm sizes: small (< 6.4 ha), medium (6.4–16 ha), and large farms (> 16 ha). The information was collected from growers through interviews and relevant documents were collected. The questionnaires referred to the topic of "how to produce mangoes of sufficient quality for supply to the exporter".

2.4.2. Exporter

The second set of observations were made at a packing house belonging to the mango exporter. An observation form was used to evaluate the handling process. The information was analyzed to gauge the importance of processing in preventing post-harvest diseases.

*2.5. Collection of Mangoes for Disease Evaluation (Stage 5)*

At the field of production, 'Nam Dok Mai No. 4' and 'Nam Dok Mai Sithong' mangoes were sampled from 5 orchards in the Phetchabun province (30 fruits from each). The fruit was washed, dipped in 400 mg·L$^{-1}$ ethephon, and then incubated at ambient conditions (25 °C, 60–70% RH).

At the packing house, the mangoes were sampled at each step (32 fruits per step) of the mango processing line. The fruit sampled during Step 1 (material receiving), Step 2 (pedicle cutting and washing), and Step 3 (hot water dipping) was dipped in 400 mg·L$^{-1}$ ethephon. All sampled fruit was then kept at 25 °C, 60–70% RH for 9 days to allow for ripening and the development of disease symptoms was then checked. The infected fruit was counted when the fruit showed sharp spots of disease.

Percentage of disease incidence in mangoes was calculated using the following formula:

$$\% \text{ Disease incidence} = (a/b) \times 100 \tag{1}$$

When

a = number of all infected fruit in each treatment
b = number of all fruit in each treatment

*2.6. Analysis of Data (Stage 6)*

The collected data was analyzed using a SPSS program (Version 19., IBM Corporation, Armonk, NY, USA) on MS Windows and reported as frequencies and averages. The item findings were analyzed using the SCOR model and the critical control points of post-harvest disease infection during the supply chain of 'Nam Dok Mai' mango was identified by mapping between the material and information flows.

## 3. Results and Discussion

*3.1. Supply Chains of 'Nam Dok Mai' Mango Exported from Thailand to Japan*

Mango supply chains comprised material flow and information flow. The material flow of mango was separated into three parts: production, cooperation, and distribution (Figure 2) The production aspect was explored as part of the preharvest process. Growers were important players here in terms of using tools of propagation, fertilizers, pesticides, bagging, and labor. The cooperation aspect included co-operators and an exporter. The latter plays an important role in the packing house's mango processing line in ensuring adherence to the requirements of the export destination. The exporter buys mango materials directly from growers or indirectly from co-operators. The distribution aspect included destinations in international and domestic markets. The mangoes failing to meet exporting standards are distributed in domestic markets.

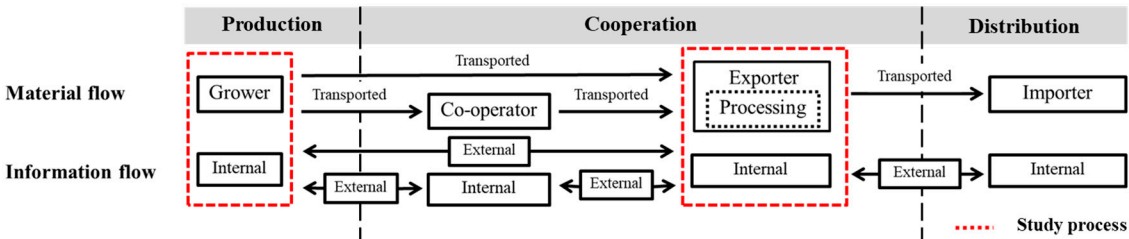

**Figure 2.** Supply chain management of Thai mangoes for export. The boxes highlighted with the red dashed lines indicate the processes focused on in this study.

The information flow consisted of internal and external information. Internal information is the confidential or proprietary data of a company used within its own span of operations to track ingredients/products, whereas external information refers to the data exchange and business processes taking place between trading partners. However, the study of the mango production process for export to Japan focused on the activities and information flows of growers and the exporter.

### 3.2. Mango Process Mapping and Disease Risk Assessment for Farms

The general information obtained from interviews with 44 mango growers regarding their actual activities of mango production in the field is shown in Table 1. All the growers stated that they have received GAP certification for their mango production and most of them have been undertaking mango production as per GAP for 4–8 years. The flow of material from planting to fruit harvesting following the Thai GAP is shown in Figure 3.

**Table 1.** General information regarding interviewed growers with different farm sizes followed by the supply chain operation reference (SCOR) model.

| SCOR Item | Descriptor | Percentage of Growers | | |
|---|---|---|---|---|
| | | **Small** | **Medium** | **Large** |
| 1. Sex | - Male | 75 | 75 | 76.9 |
| | - Female | 25 | 25 | 23.1 |
| 2. Age | - < 30 y | - | - | - |
| | - 31–50 y | 50.1 | 81.2 | 76.9 |
| | - > 51 y | 49.9 | 18.8 | 23.1 |
| 3. Education | - Primary school | 43.8 | 31.3 | 46.2 |
| | - High school | 43.7 | 43.6 | 23 |
| | - Vocational school | 0 | 18.8 | 15.4 |
| | - Bachelor's degree | 12.5 | 6.3 | 15.4 |
| 4. Experience in mango growing | - < 5 y | 6.2 | 0 | 7.7 |
| | - 6–10 y | 50 | 25 | 15.3 |
| | - 11–15 y | 18.8 | 43.8 | 30.8 |
| | - > 15 y | 25 | 31.2 | 46.2 |
| 5. GAP certification | | 100 | 100 | 100 |
| 6. GAP experience | - < 3 y | 25 | 18.8 | 30.8 |
| | - 4–8 y | 56.2 | 75 | 15.4 |
| | - > 8 y | 18.8 | 6.2 | 53.8 |
| 7. Purchase Channel | - To collector at orchard | 0 | 0 | 6.7 |
| | - To exporter at orchard | 37.5 | 43.8 | 53.8 |
| | - Transported to collecting group | 62.5 | 56.2 | 46.2 |

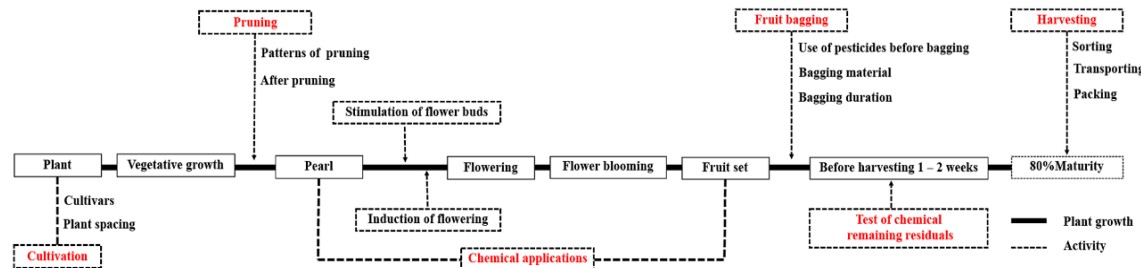

**Figure 3.** 'Nam Dok Mai' mango production process followed by growers. The risk points of post-harvest disease infection are indicated in dash line boxes with red letters.

(a) *Cultivation*: Mango growers grew both sub-cultivars of 'Nam Dok Mai' mango, namely 'Nam Dok Mai' and 'Nam Dok Mai', for 80% of total planted mangoes while the remaining 20% were other varieties (Table 2). This implies that 'Nam Dok Mai' mango is the most favored one, although the production of mango for export has several compulsory requirements including GAP certification, the limitation of chemical residues, and the absence of defects (physical damage, insect and disease symptoms) and disorders. Thus, the production must follow good procedures of cultivation to produce high quality products. The typical price of 'Nam Dok Mai' mango in terms of the export standard is approximately 3–5 times higher than other cultivars, leading to high levels of interest from growers. However, in terms of sub-cultivars, 'Nam Dok Mai Sithong' mango has been promoted to replace 'Nam Dok Mai No. 4' mango for export mainly due to its tolerance against anthracnose diseases [15]. The former is a new selection showing bright yellow fruit whereas the latter is an older cultivar with very good flavor. From our observation of mangoes from 5 different orchards in the Phetchabun province, it was seen that the percentage of infection for ripe 'Nam Dok Mai No. 4' mangoes was 73.3% while it was only 52.7% for ripe 'Nam Dok Mai Sithong' mangoes on day 11 of incubation (Table 3). This confirms the varying effect of the disease of the cultivars. Interestingly, there were 23–40% of 'Nam Dok Mai No. 4'plants still planted in the farms, which were processed for export. This means that mango growers were not aware of the crucial factor of anthracnose disease infection. This is the first risk point in the material flow that could increase post-harvest disease incidence at the destination or during retail display.

**Table 2.** Portions of mango cultivars grown in different farm sizes.

| Mango Cultivar | Percentage of Plants | | |
|---|---|---|---|
| | Small | Medium | Large |
| Nam Dok Mai No. 4 | 22.6 | 39.4 | 35.5 |
| Nam Dok Mai Sithong | 56.9 | 44.1 | 42.6 |
| Other cultivars | 20.5 | 16.5 | 21.9 |

**Table 3.** Disease incidence (%) (*n* = 30) of 'Nam Dok Mai No. 4' and 'Nam Dok Mai Sithong' mangoes cultivated in 5 orchards in the Phetchabun province during storage at ambient conditions (25 °C, 60–70% RH).

| Location | Days after Storage | | | | | |
|---|---|---|---|---|---|---|
| | 1 | 3 | 5 | 7 | 9 | 11 |
| Orchard 1 | | | | | | |
| Nam Dok Mai Sithong | 0 | 0 | 0 | 10 | 10 | 40 |
| Nam Dok Mai No. 4 | 0 | 0 | 0 | 0 | 53.3 | 86.7 |
| Orchard 2 | | | | | | |
| Nam Dok Mai Sithong | 0 | 0 | 0 | 30 | 30 | 40 |
| Nam Dok Mai No. 4 | 0 | 0 | 0 | 0 | 80 | 100 |

**Table 3.** *Cont.*

| Location | Days after Storage | | | | | |
|---|---|---|---|---|---|---|
| | 1 | 3 | 5 | 7 | 9 | 11 |
| Orchard 3 | | | | | | |
| Nam Dok Mai Sithong | 0 | 0 | 0 | 20 | 63.3 | 90 |
| Nam Dok Mai No. 4 | 0 | 0 | 0 | 13.3 | 26.7 | 70 |
| Orchard 4 | | | | | | |
| Nam Dok Mai Sithong | 0 | 0 | 0 | 10 | 10 | 10 |
| Nam Dok Mai No. 4 | 0 | 0 | 0 | 0 | 0 | 33.3 |
| Orchard 5 | | | | | | |
| Nam Dok Mai Sithong | 0 | 0 | 0 | 30 | 70 | 83.3 |
| Nam Dok Mai No. 4 | 0 | 0 | 0 | 43.3 | 50 | 76.7 |
| Total | 0 | 0 | 0 | 15.7 | 39.3 | 63 |

In addition, this study found that the growers used a planting space between $4 \times 4$ or $4 \times 6$ m$^2$ for mango trees aged between 1–8 years and $8 \times 8$ m$^2$ for older trees aged above 15 years. This suggests that growers used close spacing systems for the new mango tree generation to obtain a high yield. Gaikwad et al. (2017) [16] reported that plant space affects the fruit quality and yield; the mango variety Kesar planted with a spacing of $5 \times 5$ m$^2$ resulted in a high incidence of mango hoppers/inflorescence and powdery mildew/inflorescence compared to $10 \times 10$ m$^2$ spacing. However, the spacing of $5 \times 5$ m$^2$ was recommended due to the higher yield obtained. In addition, Ansari et al. (2018) [17] reported a spacing of $5 \times 5$ m$^2$ in combination with branch pruning, stating that the spacing effect was non-significant to the yield. Recently, growers have been pruning mango trees to stay below 2 m in height. Pruning the branches allows light to pass through the trunk, which decreases the accumulation of the anthracnose disease. The growers do not monitor for the infectious anthracnose disease during this process.

(b) *Branch pruning*: Pruning is a necessary first step in the management of flowering for the next season. It ensures not only a uniform flush of tree growth throughout the canopy but also the removal of the previous season's flowering and fruiting panicles. In particular, dead or diseased wood is removed to decrease accumulated disease. After tip pruning, the diseased wood must be removed from the orchard and destroyed (GAP guide). All growers follow the branch pruning program strictly. However, 93.3% of growers did not remove tips and 66.3% did not remove fruit rot from the mango trees after pruning (Table 4) mainly due to a lack of labor. The tips were crumbed and propped under the tree instead. The diseases thus accumulate in the production field year after year and it becomes difficult to clean or eliminate the diseases during the next production year. This is the second crucial step for field production management in terms of induction risks of post-harvest diseases.

(c) *Chemical applications*: Chemicals used to prevent diseases infections among mango trees and fruit were mostly applied during vegetative flushing (between the leaf pearl and fruit set stages) prior to fruit bagging. The fungicides used by mango growers included Prochloraz, Propinep, Mancozeb, and Azoxystrobin. Although the GAP guide suggests that chemicals be used when diseases or insects appear, chemicals were ordinarily used every 10–14 days as a precaution by most growers (Table 5). The interviews showed that more than 40% of growers did not wait until they found anthracnose disease damage but applied fungicides every 10 days. This implies that this process is important to prevent disease infection and accumulation. Nevertheless, excessive applications of chemicals increase the capital required for mango production.

(d) *Fruit bagging*: This step is essential to protect fruit skin not only from physical damage but also from insect invasion and disease infection [18]. Furthermore, for 'Nam Dok Mai No. 4' mango, this process induces the yellow skin color [19]. All growers use the Shun Fong® bag (laminated bag with carbon paper inside and brown paper outside) for mango bagging for export.

The appropriate duration to start bagging mangoes was between 45–55 days after flowering (DAF) for 'Nam Dok Mai No. 4' mango and between 60–70 DAF for 'Nam Dok Mai Sithong' mango. The GAP guide suggests bagging mangoes during 40–50 DAF in general. Of the growers, 80% followed the bagging process mostly 1 day after pesticide application (Table 5). After fruit bagging, growers significantly reduced chemical use and fungicides were applied only when disease damage was found. This is another crucial step to reduce fruit bruising and post-harvest disease infection. However, it is a cause for concern that above 30% of growers reused bagging material more than 2 times, which can lead to an accumulation of diseases.

(e) *Chemical residual checking*: One or two weeks before harvesting, mangoes were randomly sampled by exporter staff to check for chemical residues. Chlorpyrifos must remain less than $0.05$ mg·L$^{-1}$ on fruit for it to qualify for Japanese markets. Once chemical residue levels were approved, the exporter staff then scheduled the harvest date of mangoes based on 80% of maturation time.

(f) *Harvesting*: Mangoes in the Shun Fong$^{®}$ bag were harvested in the morning. The harvested fruit was transported to the grower's packing house or to the co-operator's premises and the paper bags were removed. If the bags were removed in the field, mangoes usually displayed defects such as scratches during transportation, which lead to anthracnose infection [20]. Mangoes without defects were sorted by the exporter staff. The selected fruit was covered with a foam net before being placed in plastic baskets and transported to the exporter's packing house. Mangoes not meeting the standard were, on the other hand, sold to domestic markets.

**Table 4.** Practices in mango production fields with different farm sizes followed by the supply chain operation reference (SCOR) model.

| SCOR Item | Descriptor | Percentage of Growers | | |
|---|---|---|---|---|
| | | **Small** | **Medium** | **Large** |
| 1. Branch trimming | | 100 | 100 | 100 |
| 2. Trimmed branch management | - moved from the field | 0 | 12.5 | 7.7 |
| | - kept in the field | 100 | 87.5 | 92.3 |
| 3. Fallen fruit management | - moved from the field | 31.3 | 31.3 | 38.5 |
| | - kept in the field | 68.7 | 68.7 | 61.5 |
| 4. Irrigation | - practiced | 18.8 | 0 | 30.8 |
| | - not practiced | 81.2 | 100 | 69.2 |
| 5. Source of water | - ground water | 68.8 | 43.7 | 46.2 |
| | - irrigation canal | 6.2 | 18.8 | 7.7 |
| | - both ground water and irrigation canal | 6.2 | 37.5 | 15.3 |
| | - not practiced | 18.8 | 0 | 30.8 |
| 6. Fruit fly control | - chemical spray | 93.8 | 100 | 92.3 |
| | - insect trap | 6.3 | 0 | 7.7 |

Mango growers were not aware of anthracnose infection because its symptoms cannot be detected by visual appearance during the unripe stages. As a latent infection, *Colletotrichum gloeosporioides* grows sharply and symptoms manifest when the mango ripens [6,21]. Anthracnose infection was not the main focus of 39% of growers when they applied fungicides to mango trees every 14 days (Table 5). In addition, interviews with mango growers concerning traceability (Table 6) found that 98% of the growers did not realize the importance of material traceability. For example, they could not recognize and identify a particular mango material lot of a sub-field from other production areas because of a lagging tracking system. Mango growers only recorded data such as applied chemical date, fruit bagging date, types of chemicals used, and rate of chemicals applied when the GAP certification was renewed. Only a few mango growers (11%) always recorded activities because they used the recorded information to calculate the income and expenditure in farms. Although all the mango growers possess GAP certification, internal information was not recorded by the growers.

**Table 5.** Practices in mango production fields with different farm sizes followed by the supply chain operation reference (SCOR) model.

| SCOR Item | Descriptor | Percentage of Growers | | |
|---|---|---|---|---|
| | | **Small** | **Medium** | **Large** |
| 1. Frequency of pesticide application | | | | |
| | - every 7 days | - | - | - |
| | - every 10 days | 62.5 | 43.8 | 84.6 |
| | - every 14 days | 37.5 | 56.2 | 15.4 |
| 2. Pesticide application before fruit bagging | | | | |
| | - less than 1 day | 68.8 | 68.8 | 61.5 |
| | - 1 day | 18.8 | 25 | 38.5 |
| | - 2 days | 12.4 | 6.2 | 0 |
| 3. Recycling of bagging materials | | | | |
| | - Not reused | 6.3 | 6.3 | 0 |
| | - 2 times | 56.3 | 56.3 | 69.2 |
| | - 3 times | 37.4 | 37.4 | 30.8 |

**Table 6.** Traceability management in mango orchards.

| SCOR Item | Descriptor | Percentage of Growers |
|---|---|---|
| Do you know "Trace-back System"? | -no<br>-yes | 97.93<br>2.07 |
| Can you trace back your product to your field? | -unable<br>-able | 71.97<br>28.03 |
| Field management recording | -always<br>-only when applying for the renewal of GAP | 10.87<br>89.13 |

The general problems affecting mango production are shown in Table 7. Natural disasters (especially climate change), labor shortage, and disease and insect invasion were major concerns for the mango supply chain. Surprisingly, chemical residues were not detected at all for mangoes sent to the exporter's packing house. The main reason for this was strict monitoring by exporter staff prior to harvesting.

**Table 7.** Problems faced by mango growers during production supply chain.

| Problems | Percentage of Growers | | | |
|---|---|---|---|---|
| | **Small** | **Medium** | **Large** | **Average** |
| 1. Uncertainty of price | 12.5 | 31.3 | 38.5 | 26.7 |
| 2. Low quality of fruit | 12.5 | 37.5 | 15.4 | 22.2 |
| 3. No irrigation system | 25 | 43.8 | 30 | 33.3 |
| 4. Natural disasters | 100 | 81.3 | 84.6 | 88.9 |
| 5. Diseases and insects | 56.3 | 68.8 | 38.5 | 55.6 |
| 6. Labor shortage | 68.8 | 62.5 | 76.9 | 68.9 |
| 7. Chemical residue remaining | 0 | 0 | 0 | 0 |

*3.3. Mango Process Mapping in the Packing House of the Exporter*

The result from the observations and interview with the mango exporter is elucidated in Figure 4.

(a) *Mango grower's orchard:* 'Nam Dok Mai' mangoes are produced in Thailand all year round, starting in the east during November–April, Central Thailand during January–May,

the north during May–June, the northeast during August–October, and the west during September–December. The exporter collected mangoes using two ways: direct transportation from growers and collection from co-operators. However, whether direct or indirect ways were used, the growers' information and relevant codes were officially issued by the exporter staff. On the harvest date, the exporter staff were ready at farms to select, grade (280–450 g), pack, and transport mangoes to the packing house of the exporter. Mangoes were transported at night and generally arrived at the packing house early in the morning depending on the distance.

(b)　*Warehouse of exporter*: Mangoes from different growers were separated using various colors of plastic containers, and paper notes with the name of the grower and received date were attached. The post-harvest handling of mango followed the method of "first in, first out" (Step 1). The process of mango handling began with cutting the pedicle and washing the fruit in a 200 mg·L$^{-1}$ chlorine solution (Step 2). Mangoes were dipped in hot water at 50 °C for 3 min and left to cool down in the air (Step 3). Fruit was then dipped in 400 mg·L$^{-1}$ ethephon (Step 4). After air drying, mangoes were graded and treated by vapor heat to increase the pulp temperature to 47 °C for 20 min, this was done by agricultural technical officers from both Thailand and Japan (Step 5). The treated fruit was packed with 3–5 kg per box. A sticker indicating the production date, grower code, and expiry date was put on each individual fruit. Before departure to the airport, the mangoes were randomly sampled for quality checking by agricultural technical officers from both Thailand and Japan's Departments of Agriculture (DOA).

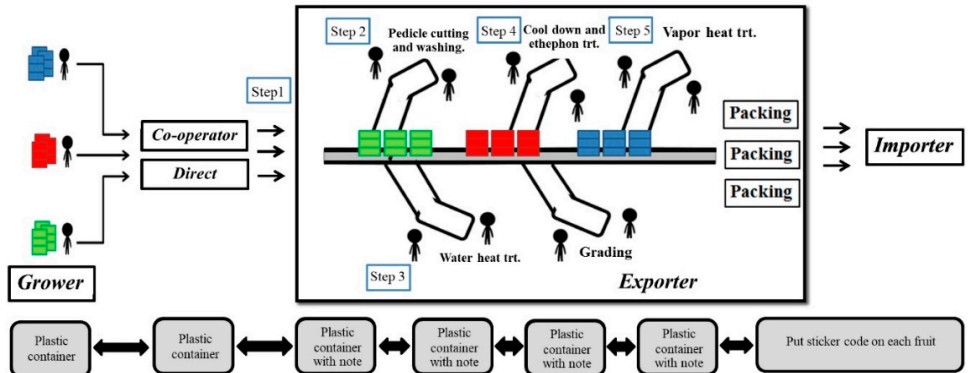

**Figure 4.** Overview of material and information flows of mango production process in the pilot export company.

The evaluation of the disease control points of 'Nam Dok Mai' mango during the post-harvest handling for Japan export was studied by separating the study into 5 steps as mentioned above. Our results showed that disease incidence in ripe mangoes were highest in steps 1 and 2 (Figure 5). Juntap (1999) [22] reported that the length of the pedicle attached to the mangoes affects disease development. When the mango was dipped in hot water at 50 °C for 3 min (Step 3), the disease incidence was 50%lower than in Step 2. Step 3 was an important step, or in other words, the so-called rate limiting step for the reduction of disease infection. This certainly confirms that hot water treatments effectively reduce disease and decay in mangoes [21–24]. Dipping the fruit into 400 mg·L$^{-1}$ ethephon (Step 4) and incubating with VHT (Step 5) showed no difference in the percentage of disease incidence and black spot areas compared to Step 3. Consequently, VHT on the processing line did not further decrease post-harvest diseases in ripe mangoes. VHTs were reported to have an effect on declining anthracnose and stem end rot in treated mangoes during ripening [25,26].

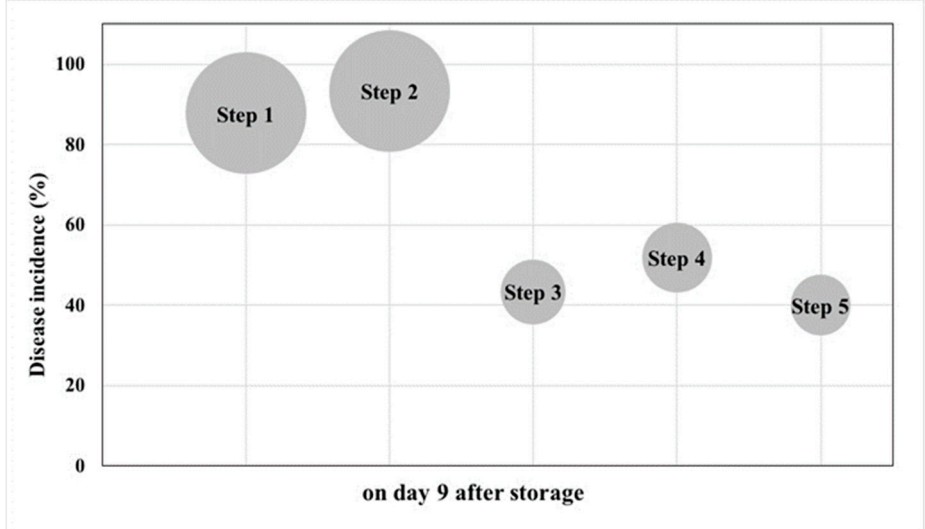

**Figure 5.** Anthracnose disease incidence and severity of mangoes from each step during handling process in the packing house of pilot export company. Size of the circle indicates the infected area on the surface of inspected mangoes.

## 4. Conclusions

There are 4 critical points affecting post-harvest disease infection which have been obtained from the study of the SCM of mango production for export to Japan, and this study examined processes ranging from pre- to post-harvest handling. In the production field, the selection of the cultivar 'Nam Dok Mai' mango was the first attempt to reduce post-harvest loss by disease infection since 'Nam Dok Mai Sithong' mango is more tolerant to anthracnose. Second, the removal of pruned branches and rotten fruit from the production area could also dramatically reduce an accumulation of diseases. Third, fruit bagging not only improves the quality of fruit but also reduces the cost of chemicals used and disease incidence. Furthermore, on the part of the exporter, the rate limiting step for a significant decrease in post-harvest disease expression is hot water treatment at 50 °C for 3 min prior to the ethephon treatment of ripening induction.

**Author Contributions:** Data curation, writing—original draft preparation, B.M.; methodology/visualization, P.P.; formal analysis, V.S.; investigation, P.B.; conceptualization/writing—review and editing/supervision, C.W.-A.; supervision/project administration, S.K.

**Funding:** This research was supported by a grant from the Thailand Research Fund (TRF) through the Royal Golden Jubilee Scholarship Ph.D. Program (Grant No. PHD/0059/2552).

**Acknowledgments:** We would like to thank the Postharvest Technology Innovation Center, Office of the Higher Education Commission, Bangkok 10400, Thailand for their equipment and facilities.

**Conflicts of Interest:** The authors declare no conflicts of interest.

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
