# Peer review of "Analysis of Critical Control Points of Post-Harvest Diseases in the Material Flow of Nam Dok Mai Mango Exported to Japan"

_agriculture, doi:10.3390/agriculture9090200_

Round 1
Reviewer 1 Report
Manuscript is well written but needs some minor english check. My comments are provided in attached pdf document.

Author Response
Thank you very much for your review. We have revised the the manuscript as major revision. We added more relevant results of Table 1, Table 2, Table 3, Table 4, Table 5, and Table 7. The results and discussion were revised mainly showing in yellow highlights.
The English writing was sent to check throroughly by a English agency.

Reviewer 2 Report
The goal of the authors would be to highlight some critical points in the postharvest and export of a specific variety of mango to Japan. However, the description of contets seems very superficial for all the points highlighted by the authors and the work, on the whole, does not seem very useful and interesting to the reader.
Also the description of the experiments conducted is not always clear, just as the processing of the data is poor and lacks critical considerations and statistical evaluation, as well as comparison with data from experiences of other studies. References are insufficient and are not recent.
The quality of English, in many parts, is poor.
Author Response
Thank you very much for your review. We have revised the the manuscript as a major revision.
Point 1: the description of contents were superficial
Response: We added more relevant results of Table 1, Table 2, Table 3, Table 4, Table 5, and Table 7. The results and discussion were revised and the additions were mainly showed in yellow highlights as the attachment.
Point2: The description of the experiments is not clear
Response: We revised the session of materials and methods which could improve the area.
Point 3: References are insufficient and are not recent.
Response: We revised some references to be updated. Many relevant references are in Thai Language so that we selected some of standard references to be cited.
Point4: English is poor.
Response: The English writing was sent to check clearly by an English agency.

Reviewer 3 Report
Dear Authors,
although the topic is potentially very interesting the article fails to give a real overview of the pitfalls in mango postharvest handling chain before export to Japan. In my opinion several relevant information are not or only partially provided. In some parts it seems that a mere review of the exsisting literature has been provided. Furthermore, English and style must be improved. As such, in my opinion the manuscript needs an extensive re-tinking and re-writing not compatible with the time of a revision
Author Response
Thank you very much for your review. We have revised the the manuscript as a major revision.
Point 1: Lack of relevant information
Response: We added more relevant results of Table 1, Table 2, Table 3, Table 4, Table 5, and Table 7. The results and discussion were revised and the additions were mainly showed in yellow highlights as the attachment.
Point2:English and style must be improved.
Response: The English writing was sent to check clearly by an English agency.

Reviewer 4 Report
Although quality is such a subjective attribute, some objective parameters are determinant for successful marketing of fresh produce. Among these, besides quality attributes closely linked to taste and flavour, visual appearance, peel disorders and even more, decay incited by microorganism are extremely important.
Failure to maintain quality and prevent decay along the marketing chain does not depend only effective postharvest treatments, but also on pre-harvest factors and on the various steps that characterize transport and handling chain.
In this study authors did a valuable job in identifying the critical steps that characterize the whole chain of mango, from the field to the packing house and the handling process.
The results of this study are valuable and of great importance to improve the marketing conditions of mango and to reduce quality loss and decay, mainly caused by two filamentous fungi: Colletotrichum gloeosporioides and Lasiodiplodia theobromae, along the distribution chain.
My feeling is that, after improving the English style, this manuscript is worth for publication in this Journal.

Author Response

(The authors gave the same response as above.)

Round 2
Reviewer 3 Report
The manuscript improved sufficiently for publication